# COVID-19 Vaccines and Restrictions: Concerns and Opinions among Individuals in Saudi Arabia

**DOI:** 10.3390/healthcare10050816

**Published:** 2022-04-28

**Authors:** Abdulkarim M. Meraya, Riyadh M. Salami, Saad S. Alqahtani, Osama A. Madkhali, Abdulrahman M. Hijri, Fouad A. Qassadi, Ayman M. Albarrati

**Affiliations:** 1Department of Pharmacy Practice, College of Pharmacy, Jazan University, Jazan P.O. Box 114-45124, Saudi Arabia; ssalqahtani@jazanu.edu.sa; 2Pharmacy Practice Research Unit, College of Pharmacy, Jazan University, Jazan P.O. Box 114-45124, Saudi Arabia; riyadhs172@gmail.com (R.M.S.); dhoomjzn@gmail.com (A.M.H.); afuad0272@gmail.com (F.A.Q.); 3Department of Pharmaceutics, College of Pharmacy, Jazan University, Jazan P.O. Box 114-45124, Saudi Arabia; omadkhali@jazanu.edu.sa; 4Medicine College, Jazan University, Jazan P.O. Box 114-45124, Saudi Arabia; albratyayman@gmail.com

**Keywords:** COVID-19, vaccine, worry, Saudi Arabia

## Abstract

(1) Background: Recent studies in Saudi Arabia have indicated that a small proportion of the population is hesitant to receive COVID-19 vaccines due to uncertainty about their safety. The objective of this study was therefore to examine concerns about COVID-19 vaccines in Saudi Arabia; (2) Methods: This cross-sectional study surveyed Saudi residents aged 14 years and older. The online questionnaire consisted of the following sections: (1) demographics; (2) knowledge about COVID-19 vaccines and sources of information; (3) COVID-19 vaccines worry scale; and (4) opinions about restrictions placed on unvaccinated individuals in Saudi Arabia. An adjusted regression model was computed to examine the relationships between demographic factors and worry about COVID-19 vaccines. All analyses were stratified by age, with those aged 19 years and above considered adults and those aged younger than 19 years considered as adolescents; (3) Results: A total of 1002 respondents completed the survey. Of the study sample, 870 were aged ≥19 years and 132 were aged <19 years. Of the adults in the study sample, 52% either agreed or strongly agreed with the statement, “I am worried about the potential side effects of COVID-19 vaccinations in children”. Among adults, females demonstrated higher levels of worry about COVID-19 vaccines than males (β = 1.142; *p* = 0.004) in the adjusted analyses. A high percentage of the participants either disagreed or strongly disagreed with allowing unvaccinated individuals to enter malls, schools, universities or to live freely without restrictions; (4) Conclusions: A high proportion of individuals in Saudi Arabia are concerned about possible side effects of COVID-19 vaccines, and many believe that unvaccinated individuals should not be restricted from participating in public life. It is therefore crucial to provide easily accessible information on the safety of COVID-19 vaccines in order to accelerate vaccination and minimize hesitancy regarding any future vaccinations that may be necessary.

## 1. Introduction

Coronavirus disease 2019 (COVID-19) was first reported in China and spread to most countries in the world [1]. To date, billions of people around the world have been affected with the disease and its complications [1]. COVID-19 vaccines have helped in reducing the number of COVID-19 cases and related deaths worldwide [2,3]. In the United States, the rates of COVID-19 cases, hospitalizations, and deaths were lower in fully vaccinated individuals [2]. In Saudi Arabia, the number of COVID-19 cases and deaths has decreased dramatically since the introduction of the first approved vaccine on 17 December 2020 [4]. On 28 June 2021, there were 9609 confirmed COVID-19 cases with 718 weekly increases. As of 24 September 2021, there were 576 confirmed cases with 222 weekly decreases [4,5], and 18,130,687 individuals had been fully vaccinated. Although numerous studies and reports have indicated that COVID-19 vaccines are efficacious and safe among different subpopulations [6,7,8,9], a small proportion of the Saudi population is hesitant to receive vaccination [10,11].

In May of 2020, before the introduction of the COVID-19 vaccine, 55% of 3101 surveyed adults living in Saudi Arabia were hesitant to receive COVID-19 vaccination, and 80% reported that they were concerned about the vaccine’s side effects [11]. A study conducted between 25 December 2020 and 15 February 2021 on a sample of 3084 Saudi residents found that 47% had either refused vaccination or were not sure if they were willing to be vaccinated [10]. Additionally, another study with a sample of 1387 individuals found that 43% had either a probable or definite negative intent to receive COVID-19 vaccination [12]. Most of these studies reported fear of vaccine-related side effects as a reason for their hesitancy. Other factors negatively associated with COVID-19 vaccination intent were concerns about effectiveness and long-term consequences [12]. It is important to minimize this hesitancy in all population groups, as vaccination in Saudi Arabia is currently recommended for adolescents and pregnant women and will soon be recommended for children.

Therefore, the main objective of this study was to examine concerns about COVID-19 vaccines in Saudi Arabia. Other objectives were to assess knowledge about COVID-19 vaccines, to identify common sources of information about the vaccine, and to determine opinions about COVID-19-related restrictions placed on unvaccinated individuals.

## 2. Material and Methods

### 2.1. Study Population and Recruitment Procedure

A survey-based cross-sectional study was conducted to examine worry about COVID-19 vaccines and opinions about COVID-19-related restrictions for unvaccinated individuals in Saudi Arabia. Following the IRB approval from Jazan University (REC42/1/6; 10 June 2021), cognitive interviews were conducted among 10 participants to assess the face validity of the questionnaire. The eligible participants were Saudi citizens aged 14 years and older. The study was conducted from 22 July 2021 to 1 September 2021. A snowball sampling method was used to recruit participants. Initially, 30 participants were selected based on age, education level, and region of residence, and these participants were then asked to forward the questionnaire to acquaintances whom they considered suitable for the survey. The participants were asked to fill out an online consent form at the start of the questionnaire and were informed that they could withdraw from the questionnaire at any time. The questionnaire was distributed online via Google forms, and the web link was sent via email and circulated to various social media platforms such as Twitter, WhatsApp, and others to increase the visibility of the survey. Of the 1173 individuals who started the survey, 1002 completed it (870 adults and 132 adolescents).

### 2.2. Measures

The questionnaire consisted of the following sections: (1) demographics; (2) knowledge about COVID-19 vaccines and sources of information; (3) COVID-19 vaccines worry scale; and (4) opinions about COVID-19-related restrictions on unvaccinated individuals.

#### 2.2.1. Demographics

Demographic variables included age, sex (male and female), marital status (single and married), education level (≤middle school, high school, diploma, BSc, >BSc), region of residence (Al Baha, Al Jawf, Al Madinah, Al Qassim, Ásir, Al Sharqiyah, Hai’l, Jazan, Makkah, Najran, Northern Borders, Riyadh, and Tabuk) and area (city and village).

#### 2.2.2. Knowledge about COVID-19 Vaccines and Sources of Information

Participants’ knowledge about COVID-19 vaccines was assessed by asking the following questions: (i) How did you learn that the new COVID-19 vaccines were licensed for use in Saudi Arabia? (ii) What is your primary source of information about COVID-19 vaccines? (iii) Which COVID-19 vaccines are available in Saudi Arabia? (iv) Are the new COVID-19 vaccines 100% effective against infection? (v) Are the new COVID-19 vaccines regarded as safe? (vi) Do the newly developed COVID-19 vaccines reduce risk of disease? (vii) Can precautions such as wearing a mask and keeping a safe distance be ignored after receiving the COVID-19 vaccine?

#### 2.2.3. COVID-19 Vaccines Worry Scale

Worry about COVID-19 vaccines was measured using a 5-item scale, which included the following statements: (i) I am worried about the short-term side effects of COVID-19 vaccines; (ii) I am worried about the long-term side effects of COVID-19 vaccines; (iii) I am worried about the potential side effects of COVID-19 vaccines in children.; (iv) I am worried about the potential side effects of COVID-19 vaccines in pregnant women; (v) I am worried about the future consequences of COVID-19 vaccines in the population. Participants responded to these questions using a Likert-type scale ranging from “strongly disagree” to “strongly agree”.

#### 2.2.4. Opinions about COVID-19-Related Restrictions for Unvaccinated Individuals

Participants’ opinions about COVID-19-related restrictions for unvaccinated individuals were assessed using a 4-item scale that included the following statements: (i) I think it is preferable to make the COVID-19 vaccine optional; (ii) I think it is preferable to allow unvaccinated people to enter malls; (iii) I think it is preferable to allow unvaccinated students to attend schools/universities; (iv) I think it is preferable to allow unvaccinated people to live freely. Participants responded to these questions using a Likert-type scale ranging from “strongly disagree” to “strongly agree”.

### 2.3. Statistical Analysis

Frequency and percentages were reported for categorical variables. Means and standard deviations were reported for continuous variables. Separate validity and reliability tests were conducted for each scale to select final items for COVID-19 vaccine worry scale and opinions about COVID-19-related restrictions for unvaccinated individuals. All analyses were stratified by age (adults ≥ 19 years and adolescents < 19 years). A multivariable ordinary least regression model was applied to assess the relationship between demographic variables and COVID-19 worry among adults only. All statistical analyses were performed using Stata 16.0 (Stata Corp LP, College Station, TX, USA).

## 3. Results

### 3.1. Demographics, Knowledge about COVID-19, and Sources of Information

The study sample consisted of 1002 Saudi citizens aged 14 years and older. Of the study sample, 870 were aged ≥19 years and 132 were aged <19 years. Among those aged 19 or older, most were male (56%), single (63.7%), and had a Bachelor’s degree (59.4%). Additionally, 45.9% reported that they first heard about COVID-19 vaccines from social media, while 60.6% identified the Saudi Ministry of Health (MOH) as their primary source of information about COVID-19 vaccines. Among adolescents (<19 years), most were female (75%) and had finished high school (78.8%). Furthermore, 43.9% reported that they first heard about COVID-19 vaccines from social media, while 65.2% reported the Saudi MOH as the primary source of information. Table 1 displays detailed information on demographics, knowledge about COVID-19 vaccines, and sources of information.

### 3.2. COVID-19 Vaccines Worry Scale

Based on the factor analysis with oblique rotation (Promax), the worry scale consisted of a single underlying factor. The Cronbach’s alpha for the scale was 0.91. The mean score for COVID-19 vaccine worry among adults (*n* = 870) was 12.2 (*SD* = 5.7). Of the adults in the study sample, 57.8% either agreed or strongly agreed with the statement, “I am worried about the potential side effects of COVID-19 vaccinations in pregnant women”; 52% either agreed or strongly agreed with the statement, “I am worried about the potential side effects of COVID-19 vaccinations in children”; and 48% either agreed or strongly agreed with the statement, “I am worried about the future consequences of COVID-19 vaccines on the population”. With respect to adolescents, the mean score for COVID-19 vaccine worry was 11.9 (*SD* = 5.4). Among the adolescents, 53.3% either agreed or strongly agreed with the statement, “I am worried about the potential side effects of COVID-19 vaccinations in children”; 55.5% either agreed or strongly agreed with the statement, “I am worried about the potential side effects of COVID-19 vaccinations in pregnant women”; and 53% either agreed or strongly agreed with the statement, “I am worried about the long-term side effects of COVID-19 vaccinations”. Figure 1: Worry scale regarding COVID-19 vaccines shows the results regarding adults’ and adolescents’ worry about COVID-19 vaccines.

### 3.3. Demographics and Worry about COVID-19 Vaccines among Adults

Table 2 shows parameter estimates of the demographic variables from an ordinary least squares regression on worry about COVID-19 vaccines. In the adjusted analyses, there was a significant relationship between sex and worry about COVID-19 vaccines, with females demonstrating higher levels of worry about COVID-19 vaccines than males (β = 1.142; *p* = 0.004). Additionally, married individuals demonstrated more worry about COVID-19 vaccines than singles (β = 1.553; *p* = 0.003). With regard to education, only those with a diploma showed significantly higher levels of worry about COVID-19 vaccines than those with a middle school education or less (β = 3.572; *p* = 0.004). Nevertheless, it should be noted that the coefficients for all education levels were positive. 

### 3.4. Opinions about COVID-19-Related Restrictions for Unvaccinated Individuals

Figure 2 Participants’ opinion about COVID-19 restrictions for unvaccinated individuals displays participants’ opinions about COVID-19-related restrictions for unvaccinated individuals, stratified by age into adults and adolescents. Of the adults, 50% either agreed or strongly agreed with the statement, “I think it is preferable to make the COVID-19 vaccine optional”, but 47% either disagreed or strongly disagreed with the statement, “I think it is preferable to allow unvaccinated students to attend schools/universities”. Furthermore, 44% reported that they disagreed or strongly disagreed with the statements, “I think it is preferable to allow unvaccinated people to enter malls”, and, “I think that it is preferable to allow unvaccinated people to live freely”. Likewise, a large number (49%) of the adolescents either agreed or strongly agreed with the statement, “I think it is preferable to make the COVID-19 vaccine optional”, although 45% either disagreed or strongly disagreed with the statements, “I think it is preferable to allow unvaccinated students to attend schools/universities”, and, “I think that it is preferable to allow unvaccinated people to live freely”.

## 4. Discussion

This study examined worry about COVID-19 vaccines and opinions about COVID-19-related restrictions for unvaccinated individuals among a sample population in Saudi Arabia. A high proportion of adults and adolescents in the study sample were worried about the possible effects of COVID-19 vaccines on children and pregnant women. Pregnant women are at higher risk of experiencing severe COVID-19 illness and of requiring hospitalization [13,14]. Numerous studies and reports have indicated that COVID-19 vaccines are safe for use in pregnant women and that the benefits outweigh the risks [6,7,8,9], but pro-vaccination attitude among pregnant women varies by country, and in some countries (including the United States, Australia, Russia, and Turkey) is below 45% [15,16]. Furthermore, COVID-19 vaccines are authorized for use in adolescents (12–17 years) in most countries, and are regarded as safe [17]. Some recent reports have indicated that COVID-19 vaccines are safe for children younger than 12 years old, and that they could be administered to them in the near future [18]. To increase COVID-19 vaccination rates in Saudi Arabia and reduce hesitancy to receive any future vaccinations that may be necessary, it is crucial to provide easily accessible information in simple Arabic on the safety of COVID-19 vaccines for children and pregnant women.

In the sample, females were generally more worried about COVID-19 vaccines than males. Females in Saudi Arabia experienced higher rates of depression, general anxiety, and other forms of psychological distress than males during the COVID-19 pandemic [19,20], and mothers and pregnant women reported being distressed by fear of their children contracting COVID-19 [21]. It therefore appears important to address pandemic-related mental health concerns among Saudi women, in addition to implementing measures to ensure physical health. To this end, the Saudi Ministry of Health, along with healthcare professionals and experts, must work to illustrate the safety of COVID-19 vaccines in women and children and disseminate that information through relevant channels and local institutions.

A high percentage of the participants either disagreed or strongly disagreed that unvaccinated individuals should be allowed to enter malls, schools, universities, or to live freely without restrictions, a stance that is consistent with the restrictions applied by the Saudi government. Most of the participants were knowledgeable about COVID-19 vaccines and reported receiving most of their vaccine information from the Saudi Ministry of Health. The Saudi MOH instated a daily press conference to provide the public with summaries of daily cases, deaths, and recommendations. Currently, the Saudi MOH holds periodic press conferences before any major change in policy. These practices, along with other preventive measures, have increased public confidence in both the Saudi government and the Saudi MOH. It has been recommended that health authorities focus on vaccination safety and provide training to local institutions and health care professionals, to ensure that they can convey the importance of vaccination and remain up-to-date on recommendations. 

In Saudi Arabia, COVID-19 vaccine acceptance rates vary by subpopulation. In December 2020, 50% of healthcare professionals reported their intent to receive COVID-19 vaccine [22]. Nevertheless, 48–53% of the general population were willing to receive COVID-19 vaccine between December 2020 and February 2021 [11,23]. The most reported reason for low COVID-19 vaccine acceptance was concerns about vaccine-related side effects [10,11]. A high percentage of adults and adolescents in this study sample reported that they are worried about the possible effects of COVID-19 vaccines. The Saudi MOH needs to tailor health initiatives to increase vaccine uptake by conveying transparent information to the public through authentic sources. 

This study had some limitations. The results may not be generalizable to the whole population of Saudi Arabia, as most of the participants were young and highly educated, and both sampling and response bias are possible as this was an online, questionnaire-based study. Nevertheless, the study has several strengths. To the best of the authors’ knowledge, this is the first study to measure worry about COVID-19 vaccines in Saudi Arabia. In addition, this study had a relatively large sample size that included participants from different regions within Saudi Arabia, therefore likely capturing a broad range of local healthcare and health information experiences.

## 5. Conclusions

A high proportion of individuals in Saudi Arabia are worried about COVID-19 vaccines’ side effects, especially in children and pregnant women. It is crucial to provide information in simple Arabic on the safety of COVID-19 vaccines for children and pregnant women so that COVID-19 vaccine administration can continue efficiently. Such accessible information may also help to minimize hesitancy regarding vaccinations and increase vaccinations uptake. The emergence of COVID-19 variants makes vaccinations necessary not only for unvaccinated individuals, but also for vaccinated individuals who need booster doses. 

## Figures and Tables

**Figure 1 healthcare-10-00816-f001:**
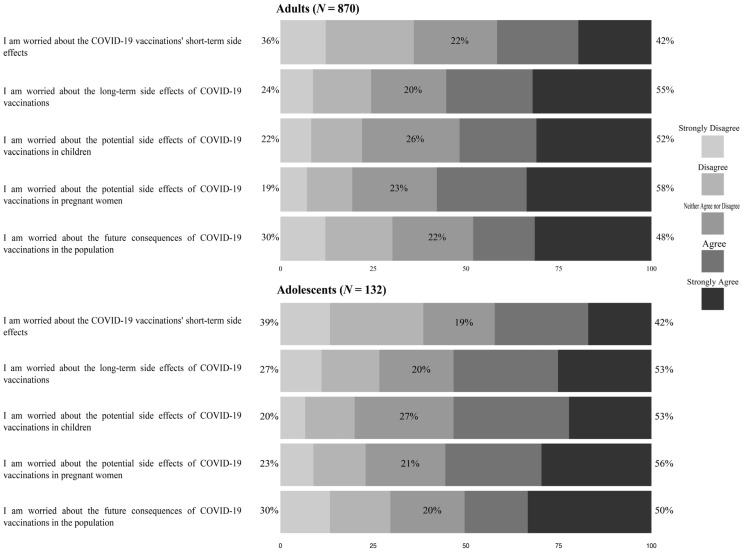
Worry scale regarding COVID-19 vaccines.

**Figure 2 healthcare-10-00816-f002:**
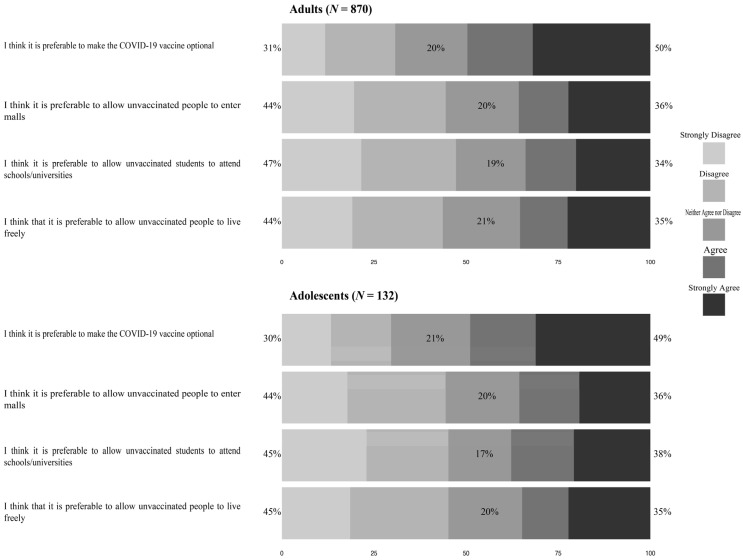
Participants’ opinion about COVID-19 restrictions for unvaccinated individuals.

**Table 1 healthcare-10-00816-t001:** Characteristics of adults and adolescents in the study sample (*N* = 1002).

Baseline Characteristic	Adults (≥19 Years) *N* = 870		Adolescents (<19 Years) *N* = 132	
	*N*	*%*	*N*	*%*
**Sex**				
Male	487	56	33	25
Female	383	44	99	75
**Marital Status**				
Single	554	63.7	130	98.5
Married	316	36.3	2	1.5
**Education**				
≤Middle	29	3.3	12	9.1
High School	191	22	104	78.8
Diploma	84	9.7	3	2.3
BSc	517	59.4	13	9.8
>BSc	49	5.6	0	0
**Were you interested in the COVID-19 vaccine?**				
No	184	21.2	35	26.5
Yes	685	78.8	97	73.5
**How did you learn that the new COVID-19 vaccines were licensed for use in Saudi Arabia?**				
Saudi Ministry of Health	315	36.2	47	35.6
TV	89	10.2	7	5.3
Social Media	399	45.9	58	43.9
Family/Friends	45	5.2	17	12.9
Other	21	2.4	3	2.3
**What is your primary source of information about COVID-19 vaccines?**				
Saudi Ministry of Health	527	60.6	86	65.2
Saudi Food and Drug Authority	28	3.2	3	2.3
Social Media	260	29.9	39	29.5
Other	54	6.2	4	3
**Have you received at least one dose of a COVID-19 vaccine?**				
No	73	8.4	33	25
Yes	795	91.6	99	75
**Are the new COVID-19 vaccines 100% effective against infection?**				
No	633	72.9	88	66.7
Yes	96	11.1	15	11.4
I don’t know	139	16	29	22
**Are the new COVID-19 vaccines considered safe?**				
No	101	11.6	11	8.3
Yes	559	64.4	78	59.1
I don’t know	208	24	43	32.6
**Do the newly developed COVID-19 vaccines reduce risk of disease?**				
No	110	12.7	22	16.7
Yes	649	74.8	89	67.4
I don’t know	109	12.6	21	15.9
**Do the new COVID-19 vaccines lessen the severity of the disease’s symptoms?**				
No	51	5.9	12	9.1
Yes	710	81.8	94	71.2
I don’t know	107	12.3	26	19.7
**Was getting vaccinated a top priority for you?**				
No	221	25.5	39	29.5
Yes	623	71.8	88	66.7
I don’t know	24	2.8	5	3.8
**Can precautions such as wearing a mask and keeping a safe distance be ignored after receiving the vaccine?**				
No	758	87.3	115	87.1
Yes	66	7.6	8	6.1
I don’t know	44	5.1	9	6.8

**Table 2 healthcare-10-00816-t002:** Parameter estimates of the demographic variables from ordinary least squares regression on worry about COVID-19 vaccines among adults (*N* = 870).

Explanatory Variable	β	95% Confidence Interval	*p*-Value
Age	0.005	(−0.049–0.06)	0.847
Sex			
Male	Reference		
Female	1.142	(0.372–1.912)	0.004
Marital Status			
Single	Reference		
Married	1.553	(0.517–2.589)	0.003
Education			
≤Middle School	Reference		
High School	1.709	(−0.628–4.047)	0.152
Diploma	3.572	(1.121–6.023)	0.004
BSc	1.881	(−0.379–4.142)	0.103
>BSc	0.837	(−1.767–3.44)	0.528
Area			
City	Reference		
Village	−0.171	(−1.071–0.729)	0.709

## Data Availability

Data are available upon reasonable request by contacting the corresponding author.

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
