# Peer review of "COVID-19 Vaccines and Restrictions: Concerns and Opinions among Individuals in Saudi Arabia"

_healthcare, 2022, doi:10.3390/healthcare10050816_

Round 1
Reviewer 1 Report
I congratulate the authors for the work done. I am grateful with the editors for the possibility of revising this manuscript. Although the quality of the manuscript is high, I would like to make some contributions that I hope will increase it and improve readers' understanding.
Introduction:
The introduction is clear and well worked.
Material and methods:
The study of the design is appropriate and well described.
Results:
The statistical analysis is correct and well described.
Discussion:
You must improve the discussion, number of references is small. Please include more references.
Change type of cite in text references and bibliography according to journal style.
Author Response
Dear Editor,
We extend and record our heartfelt gratitude to the editorial team for reviewing and sharing the comments on our manuscript. The paper has been modified in response to the insightful comments and suggestions given by the reviewers. We would like to thank the editor and the reviewers for the constructive review of our manuscript. The comments and suggestions have helped to improve the quality of the manuscript and We are greatly thankful for the opportunity to re-submit the manuscript.
We have attempted to address all the issues raised by the learned editor and reviewers in the revised manuscript.
The changes made in the manuscript are highlighted in yellow for the Reviewer #1, in blue for Reviewer #2 and green for Reviewer 3.
Reviewer #1
Comments
I congratulate the authors for the work done. I am grateful with the editors for the possibility of revising this manuscript. Although the quality of the manuscript is high, I would like to make some contributions that I hope will increase it and improve readers' understanding.
Thank you for the positive comments
Introduction:
The introduction is clear and well worked.
Material and methods:
The study of the design is appropriate and well described.
Results:
The statistical analysis is correct and well described.
Thank you.
Discussion:
You must improve the discussion, number of references is small. Please include more references.
Thank you. As per the reviewer’s suggestion, we have included the following sentences to the discussion: “In Saudi Arabia, COVID-19 vaccine acceptance rates vary by subpopulation. In December 2020, 50% of healthcare professionals reported their intent to receive COVID-19 vaccine (21). Nevertheless, 48%-53% of the general population were willing to receive COVID-19 vaccine between December 2020 and February 2021(11, 22). The most reported reason for low COVID-19 vaccine acceptance was concerns about vac-cine-related side effects (10, 11). A high percentage of adults and adolescents in this study sample reported that they are worried about the possible effects of COVID-19 vaccines. Saudi MOH needs to tailor health initiatives to increase vaccine uptake by conveying transparent information to the public through authentic sources.” Please see page 9.
Now, we have 23 references after updating introduction and discussion.
Change type of cite in text references and bibliography according to journal style.
Thank you. We have changed the reference style.
Reviewer 2 Report
Introduction: I would suggest for elaborating the introduction with more information on the hesitancy and intention towards the COVID-19 vaccinations, and also the fear on the disease. There some references which may be important for the new sentences, including one article from Saudi Arabia. Also, a scenario would be helpful from the other countries also. These articles would be useful:
- https://doi.org/10.3390/vaccines9080864
- https://doi.org/10.3390/vaccines9050416
- https://doi.org/10.1186/s12992-021-00768-3
Material and methods:
- Line 74: Please add the IRB permission number with date of approval.
- Line 84: Please elaborate the terms, "emails" and "social media". Please explain the range and type of the groups where it was spread. Was it circulated on professional Facebook groups or not, or promoted by WhatsApp etc.
- Line 105: Title of the subsection 2.2.3. should be - "Worry scale regarding COVID-19 vaccines". Same should be in other places where it should be replaced such as in line 144.
Author Response
Dear Editor,
We extend and record our heartfelt gratitude to the editorial team for reviewing and sharing the comments on our manuscript. The paper has been modified in response to the insightful comments and suggestions given by the reviewers. We would like to thank the editor and the reviewers for the constructive review of our manuscript. The comments and suggestions have helped to improve the quality of the manuscript and We are greatly thankful for the opportunity to re-submit the manuscript.
We have attempted to address all the issues raised by the learned editor and reviewers in the revised manuscript.
The changes made in the manuscript are highlighted in yellow for the Reviewer #1, in blue for Reviewer #2 and green for Reviewer 3.
Reviewer #2
Comments
Introduction: I would suggest for elaborating the introduction with more information on the hesitancy and intention towards the COVID-19 vaccinations, and also the fear on the disease. There some references which may be important for the new sentences, including one article from Saudi Arabia. Also, a scenario would be helpful from the other countries also. These articles would be useful:
https://doi.org/10.3390/vaccines9080864
https://doi.org/10.3390/vaccines9050416
https://doi.org/10.1186/s12992-021-00768-3
Thank you. We have added the following sentences to introduction “Additionally, another study with a sample of 1,387 individuals found that 43% had either a probable or definite negative intent to receive COVID-19 vaccination [12]. Most of these studies reported fear of vaccine-related side effects as a reason for their hesitancy. Other factors negatively associated with COVID-19 vaccination intent were concerns about effectiveness and long-term consequences [12]. Please, see page 2.
Material and methods:
Line 74: Please add the IRB permission number with date of approval.
Thank you. We have added the IRB permission number with date of approval (REC42/1/6; 10 June 2021).
Line 84: Please elaborate the terms, "emails" and "social media". Please explain the range and type of the groups where it was spread. Was it circulated on professional Facebook groups or not, or promoted by WhatsApp etc.
Thank you. We have modified the following sentence in the method section "The questionnaire was distributed online via Google forms, and the web link was sent via email and circulated to various social media platforms such as Twitter, WhatsApp, and others to increase the visibility of the survey."
Line 105: Title of the subsection 2.2.3. should be - "Worry scale regarding COVID-19 vaccines". Same should be in other places where it should be replaced such as in line 144
Thank you. We have accepted the above change.
Reviewer 3 Report
Manuscript ID: healthcare-1705881
Manuscript Title: COVID-19 vaccines and restrictions: concerns and opinions among individuals in Saudi Arabia
The manuscript written by Abdulkarim M. Meraya et al. focused a major social issue about COVID-19 vaccination whose content may be effective not only for Saudi Arabia but also for some regions all around the world who are quite ignorant about the necessity of vaccination, about the commencement of herd immunity to prevent the mass public health from a pandemic. The objectives, results, and interpretations have been clearly written. Methods involving the survey based cross-sectional studies are sound. The knowledge gained from this work is of global interest. However, before its publication, authors are requested to fix some minor points as appended below.
Line 26 in Abstract section may be omitted since such silly objections are quite high especially in the illiterate people because they don’t know about the essential side effects (such as, fever, pain, and other inflammation) of the vaccines which are actually in benefit for the individual. The stimulation of antibodies and the commencement of the cell mediated immunity upon vaccination are usually accompanied with such inflammation. The term “side effects” is not TRUE for the COVID-19 vaccines; rather, such effects are expected to continue for only a couple of days which are possibly due to the cytokine (including GM-CSF) and chemokine activations as well as the stimulation of the TLRs.
The Key Word “COVID-19 Vaccine Worry” in Line 37 doesn’t sound good. Authors may think of substitute words.
Page 2, Line 50: Authors should mention the exact date of the vaccination amount given.
Page 2, Line 62: What sort of “hesitancy”? Fear? Lack of knowledge on immunology? Ethical issues? Religious problems? Fundamentalist thoughts? Authors should detail this point about hesitancy of the respondents. Authors thought about these points as can be revealed in page 3.
Page 6: The Title of Figure 1 needs a good replacement. However, the Figure is clear and well presented.
Page 8, Line 207: Author should avoid the active voice. The entire manuscript should follow a passive mode write up.
In Conclusion section, authors may add a line about the SARS-CoV-2 variants for which the booster doses of vaccines are very much needed.
Author Response
Dear Editor,
We extend and record our heartfelt gratitude to the editorial team for reviewing and sharing the comments on our manuscript. The paper has been modified in response to the insightful comments and suggestions given by the reviewers. We would like to thank the editor and the reviewers for the constructive review of our manuscript. The comments and suggestions have helped to improve the quality of the manuscript and We are greatly thankful for the opportunity to re-submit the manuscript.
We have attempted to address all the issues raised by the learned editor and reviewers in the revised manuscript.
The changes made in the manuscript are highlighted in yellow for the Reviewer #1, in blue for Reviewer #2 and green for Reviewer 3.
Reviewer #3
Comments
The manuscript written by Abdulkarim M. Meraya et al. focused a major social issue about COVID-19 vaccination whose content may be effective not only for Saudi Arabia but also for some regions all around the world who are quite ignorant about the necessity of vaccination, about the commencement of herd immunity to prevent the mass public health from a pandemic. The objectives, results, and interpretations have been clearly written. Methods involving the survey based cross-sectional studies are sound. The knowledge gained from this work is of global interest. However, before its publication, authors are requested to fix some minor points as appended below.
Thank you for the positive comments.
Line 26 in Abstract section may be omitted since such silly objections are quite high especially in the illiterate people because they don’t know about the essential side effects (such as, fever, pain, and other inflammation) of the vaccines which are actually in benefit for the individual. The stimulation of antibodies and the commencement of the cell mediated immunity upon vaccination are usually accompanied with such inflammation. The term “side effects” is not TRUE for the COVID-19 vaccines; rather, such effects are expected to continue for only a couple of days which are possibly due to the cytokine (including GM-CSF) and chemokine activations as well as the stimulation of the TLRs.
Thank you. We have accepted the above change.
The Key Word “COVID-19 Vaccine Worry” in Line 37 doesn’t sound good. Authors may think of substitute words.
Thank you. We have changed the keywords to: COVID-19; vaccine; worry; Saudi Arabia.
Page 2, Line 50: Authors should mention the exact date of the vaccination amount given.
Thank you. The date is stated in line 49 "As of September 24, 2021, there were 576 confirmed cases with 222 weekly decreases [4, 5], and 18,130,687 individuals had been fully vaccinated".
Page 2, Line 62: What sort of “hesitancy”? Fear? Lack of knowledge on immunology? Ethical issues? Religious problems? Fundamentalist thoughts? Authors should detail this point about hesitancy of the respondents. Authors thought about these points as can be revealed in page 3.
Thank you. We have modified the paragraph to the following: " In May of 2020, before the introduction of the COVID-19 vaccine, 55% of 3,101 surveyed adults living in Saudi Arabia were hesitant to receive COVID-19 vaccination, and 80% reported that they were concerned about the vaccine’s side effects [11]. A study conducted between December 25, 2020 and February 15, 2021 on a sample of 3,084 Saudi residents found that 47% had either refused vaccination or were not sure if they were willing to be vaccinated [10]. Additionally, another study with a sample of 1,387 individuals found that 43% had either a probable or definite negative intent [12]. Most of these studies reported fear of vaccine-related side effects as a reason for their hesitancy. Other factors negatively associated with COVID-19 vaccination intent were concerns about effectiveness and long-term consequences [12]. It is important to minimize this hesitancy in all population groups, as vaccination in Saudi Arabia is currently recommended for adolescents and pregnant women and will soon be recommended for children." Please, see page 2.
Page 6: The Title of Figure 1 needs a good replacement. However, the Figure is clear and well presented.
Thank you. As per reviewers' 2&3 suggestions we have modified the title to "Worry scale regarding COVID-19 vaccines"
Page 8, Line 207: Author should avoid the active voice. The entire manuscript should follow a passive mode write up.
Thank you. We have accepted the above changes.
In Conclusion section, authors may add a line about the SARS-CoV-2 variants for which the booster doses of vaccines are very much needed.
“A high proportion of individuals in Saudi Arabia are worried about COVID-19 vaccines’ side effects, especially in children and pregnant women. It is crucial to provide information in simple Arabic on the safety of COVID-19 vaccines for children and pregnant women so that COVID-19 vaccine administration can continue efficiently. Such accessible information may also help to minimize hesitancy regarding vaccinations, and increase vaccinations uptake. The emergence of COVID-19 variants makes vaccinations necessary not only for unvaccinated individuals, but also for vaccinated individuals who need booster doses.” Please, see page 9.
Round 2
Reviewer 1 Report
Accept in present form.